# Neurological, Behavioral, and Pathophysiological Characterization of the Co-Occurrence of Substance Use and HIV: A Narrative Review

**DOI:** 10.3390/brainsci13101480

**Published:** 2023-10-19

**Authors:** Leah Vines, Diana Sotelo, Natasha Giddens, Peter Manza, Nora D. Volkow, Gene-Jack Wang

**Affiliations:** 1Laboratory of Neuroimaging, National Institute on Alcohol Abuse and Alcoholism, National Institutes of Health, Bethesda, MD 20892, USA; leah.vines@nih.gov (L.V.); diana.sotelo@nih.gov (D.S.); peter.manza@nih.gov (P.M.); nora.volkow@nih.gov (N.D.V.); 2Department of Psychiatry, University of Wisconsin School of Medicine and Public Health, Madison, WI 53719, USA; natashawisconsinpsychiatry@gmail.com

**Keywords:** HIV-1, Tat, gp120, psychostimulant use disorders, neuroimaging, biopsychosocial model

## Abstract

Combined antiretroviral therapy (cART) has greatly reduced the severity of HIV-associated neurocognitive disorders in people living with HIV (PLWH); however, PLWH are more likely than the general population to use drugs and suffer from substance use disorders (SUDs) and to exhibit risky behaviors that promote HIV transmission and other infections. Dopamine-boosting psychostimulants such as cocaine and methamphetamine are some of the most widely used substances among PLWH. Chronic use of these substances disrupts brain function, structure, and cognition. PLWH with SUD have poor health outcomes driven by complex interactions between biological, neurocognitive, and social factors. Here we review the effects of comorbid HIV and psychostimulant use disorders by discussing the distinct and common effects of HIV and chronic cocaine and methamphetamine use on behavioral and neurological impairments using evidence from rodent models of HIV-associated neurocognitive impairments (Tat or gp120 protein expression) and clinical studies. We also provide a biopsychosocial perspective by discussing behavioral impairment in differentially impacted social groups and proposing interventions at both patient and population levels.

## 1. Introduction

The implementation of combined antiretroviral therapy (cART) in the global healthcare system has improved health-related quality of life for people living with HIV (PLWH). Historically, HIV was considered a terminal condition and has been recharacterized as a manageable chronic condition based on cART’s ability to reduce comorbidities and prolong survival [1,2]. Before the introduction of cART, PLWH exhibited severe HIV-associated neurocognitive disorders (HANDs) and accelerated brain aging, with more prominence during the late phases of HIV progression [3]. Currently, in the cART era, the prevalence of HAND remains high, albeit with reduced severity [3,4].

Over 80% of PLWH exhibit a lifetime history of trying an illicit drug, compared to 50% in the general population [5]. Additionally, PLWH exhibit a four times higher SUD prevalence than uninfected individuals [5]. Compared to other psychiatric disorders (i.e., depression, anxiety), SUDs are the most common comorbid conditions among PLWH (40–74% vs. <50%) [6]. Cocaine and methamphetamine (meth) use disorders (CUDs and MUDs, respectively) are associated with risky sexual behavior and needle sharing, increasing the likelihood of HIV transmission [7,8,9,10]. Additionally, a higher frequency of psychostimulant use has a greater negative impact on neurocognitive functioning in PLWH than in people living without HIV, pointing to the importance of addressing this dual diagnosis [11].

Non-Hispanic Black people (NHB) and men who have sex with men (MSM) are disproportionately affected by comorbid HIV and psychostimulant use disorders [5,10,12]; however, patterns of use tend to differ based on demographics. For instance, NHB PLWH exhibit a higher prevalence of cocaine use than meth use [13]. Meanwhile, MSM tend to use meth more than cocaine [14]. Therefore, we will discuss the differential outcomes of NHB PLWH who use cocaine and MSM living with HIV who use meth. This is consistent with Dr. Wakim-Takaki’s biopsychosocial model of comorbid HIV and CUD (HIV+/CUD+), which documents complex relationships between biological, neurocognitive, and social mechanisms underlying poor outcomes in HIV+/CUD+ patients [15]. In the present review, we generalize this model to CUD and MUD since they have similar complex relationships with cellular, neurocognitive, and social mechanisms [7,8,9,10,16,17,18,19,20]. We will add to this model by reviewing evidence within the past 10 years with a central focus on studies that investigated dopamine dysregulation, applied neuroimaging methods, and differentially affected groups.

HIV and chronic psychostimulant misuse independently disrupt brain structure, function, and cognition [21,22,23,24,25,26]. The additive effects of HIV and chronic illicit psychostimulant use may be attributed in part to dysregulation of the dopamine system. Acute exposure to addictive substances transiently increases brain dopamine to supraphysiological levels, but chronic exposure attenuates dopamine signaling long-term, contributing to impairments in impulse control, learning, and memory [27]. Similarly, HIV significantly reduces dopamine synthesis, exacerbating HAND and disease progression [28,29,30].

In this review, we will address the neuropathological, behavioral, neurological, and social effects of comorbid HIV and psychostimulant use disorders (HIV+/CUD+ and HIV+/MUD+) using Dr. Wakim-Takaki’s biopsychosocial model to provide insights into the comprehensive interventions needed to address HIV+/CUD+ and HIV+/MUD+ [15]. We will first outline the neuropathogenesis of HIV and its implications for dysregulation of the dopamine reward system. We will then characterize the behavioral and biological effects of chronic psychostimulant exposure and HIV using pre-clinical and clinical evidence. In reviewing clinical evidence, we will discuss the social determinants of HIV+/CUD+ in NHB and HIV+/MUD+ in MSM, as these populations are differentially impacted by these corresponding conditions [13,14]. Finally, we will propose individual and population-level interventions per Dr. Wakim-Takaki’s biopsychosocial model [15]. To our knowledge, this is the first review within the past ten years that provides a biopsychosocial characterization and interventions for comorbid psychostimulant use disorders and HIV. It is crucial to review the literature in the context of current HIV and SUD treatments, as there are ongoing investigations and implementation of novel interventions [31,32]. This review may guide researchers, clinicians, and health policy makers in investigating and mitigating the complex effects of HIV+/CUD+ and HIV+/MUD+ through more holistic and culturally informed approaches.

## 2. Methods

Since treatment and prevention interventions for HIV have rapidly advanced over the past ten years, we searched only for studies from 2013 onward. In June 2023, we reviewed original research articles using the database PubMed and the search engine Google Scholar. Additional references were identified from previous knowledge and recursive reference searching. For the pre-clinical studies reviewed, we only included studies that model HIV-1 using Tat or gp120 protein expression in mice and rats, as these proteins are involved in HAND and HIV-induced neurotoxicity in humans [33]. For clinical studies, we only included those that had 100% of their HIV sample on antiretroviral therapy to ensure the evidence is harmonious with current medical treatments, except when discussing the effects observed in MSM HIV+/MUD+, because we identified relatively few studies that had 100% of their HIV sample on antiretroviral therapy. Meth use in this population is often a coping mechanism for psychological distress due to experiences of stigma or trauma, presenting psychosocial barriers to maintaining antiretroviral therapy [34,35,36].

## 3. Neuropathological Alterations in HIV and Evidence for Dopaminergic Dysfunction

The mechanism by which substance use contributes to HIV neuropathogenesis is not fully understood. HIV invades the central nervous system (CNS) within 1–2 weeks after the initial infection [37] through the transmigration of infected mature CD14+CD16+ monocytes across the blood-brain barrier (BBB) [38,39,40,41,42]. These cells can differentiate into long-lived macrophages and establish a viral reservoir within the CNS, infecting and activating myeloid cells, including microglia and macrophages, even with effective cART [43,44,45,46,47,48,49]. The infected and activated myeloid cells produce inflammatory cytokines and chemokines [50], leading to the recruitment and activation of additional myeloid populations and immune cells and damage to the BBB, resulting in chronic neuroinflammation [51]. Additionally, neurotoxic viral proteins such as Tat or gp120 that exacerbate the neurotoxic environment may also be released [52]. Studies also demonstrate that a small population of astrocytes harbor HIV DNA [53,54], but whether they are also viral reservoirs remains debated due to their low infectivity and replication rates [48,55,56,57]. A recent study demonstrated that although these reservoirs are low in number, HIV-infected astrocytes allow HIV to egress into the periphery, where they can repopulate the virus [58]. T-cells may also become infected and produce viruses in the CSF, which can circulate through the body and serve as a possible latent reservoir [59]. While neurons are poor viral reservoirs due to their lack of CD4 receptors, they are vulnerable to damage from chemokines, neurotoxins, and viral proteins such as Tat and gp120 [60]. Neurodegeneration is a hallmark feature of HAND, and both Tat and gp120 contribute to neurotoxicity through various mechanisms. 

Tat, trans-activator of transcription, is the first viral protein to be transcribed and translated from the integrated HIV-1 provirus and is responsible for the recruitment of positive host transcription factor P-TEFb and recognition of the 5′TAR element in HIV-1 RNA, drastically increasing the rate of viral transcription [61,62,63]. Despite cART therapy, low levels of Tat expression can result in chronic glial activation, cytokine expression, and reductions in neuronal and synaptic density [64]. Exposure of primary microglial cells to Tat results in mitochondrial dysfunction and initiates mitophagy, activating microglia and neuroinflammation [65]. Injection of Tat into the caudate putamen of rats significantly increased levels of malondialdehyde (MDA), indicating oxidative damage and induction of neuronal apoptosis [66]. It has also been shown that Tat exposure can induce autophagy, resulting in a Tat-mediated downregulation of tight junction proteins and consequently increasing vascular permeability in an in vitro model of the BBB [67]. Tat also induces a presynaptic loss in rat hippocampal neurons associated with excessive Ca^2+^ influx via NMDAR [68].

Gp120, HIV envelope glycoprotein, progresses the pathogenesis of HAND by binding to co-receptors CCR5 and CXCR4 and allowing the virus to enter host cells [69]. Gp120 evokes synaptic and behavioral deficits in vivo that mirror significant features of HAND [70]. In vitro studies have also demonstrated gp120′s role in neurotoxicity. Exposure of primary rat cortical neurons to gp120 in vitro impairs mitochondrial function by decreasing the respiratory capacity of mitochondria and disrupting mitochondrial distribution [71]. Alterations in tight junction expression, morphological changes in brain microvascular endothelial cells, and increased stress fiber formation increase BBB permeability with gp120 exposure [72]. In human monocytes, gp120 induces the production of the cytokines TNF-a and IL-10, two proteins implicated in the immunopathology of HIV-1 [73]. An increase in NMDAR-mediated excitatory postsynaptic currents measured through whole-cell patch clamp recordings on hippocampal rat brain slices may involve a molecular mechanism for gp120-induced neuronal injury [74].

Despite the success of cART, the CNS continues to serve as a viral reservoir for toxins and cytokines. Tat and gp120 proteins are expressed in transgenic rodents to investigate the neurocognitive deficits observed in HAND [70,75,76]. In non-infectious HIV-1 transgenic rats, 7 of the 9 HIV proteins (env, Tat, rev, vif, vpr, vpu, and nef) are constitutively and systemically expressed and resemble HIV-1 seropositive individuals on cART [77]. Transgenic mice with gp120 expression constitutively express gp120 in astrocytes under the control of the promoter of glial fibrillary acidic protein (GFAP) [70]. Tat and gp120 expression in transgenic rodents have been used to study the combined effects of HIV-1 and psychostimulant drug exposure. Expressing these viral proteins in transgenic rodents can help elucidate mechanisms behind the altered dopaminergic systems observed in these comorbid conditions [78,79,80]. However, since these models only express some HIV-1 viral proteins, results from these models may miss the interactive and additive effects among these proteins. 

Dopamine dysfunction has also been associated with HIV infection, and research has revealed selective damage to brain regions to which dopamine cells project, including the striatum [29,81]. While cART diminishes damage to these regions, PLWH on cART still shows striatal dysfunction, increased microglial activation, and inflammation, leading to neuronal damage [82,83,84,85,86]. Dopamine signaling has immunomodulatory effects by regulating the activation of myeloid and T-cells, the production of cytokines, transmigration, and phagocytosis [87,88,89,90], which could influence the development of HIV infection in the CNS and neurocognitive function. CD14+CD16+ monocytes, key mediators of HIV neuropathogenesis, express mRNA for all five dopamine receptors [88]. Dopamine and D1-like receptor agonists increased CD14+CD16+ cell motility, adhesion, and transmigration across the BBB in an in vitro model, suggesting that elevated extracellular dopamine in the CNS of PLWH with SUD contributes to HIV neuropathogenesis by increasing the accumulation of monocytes in dopamine-rich regions [87,88]. Elevated dopamine increases the susceptibility of macrophages to HIV [29] and spurs the production of inflammatory cytokines [91]. These effects of dopamine on immune function may promote the spread of viral infection, viral reservoirs, neuroinflammation, and neurotoxicity. Substance use is associated with increased HIV neuropathogenesis and neurocognitive decline in PLWH in the cART era [92,93,94,95], suggesting increased extracellular dopamine from substance use may play a role in the neurotoxic effects of HIV even when viral titers are very low.

Emerging evidence from pre-clinical studies suggests an HIV-mediated potentiation of drug reward [17,75,79,96,97]. These effects are theorized to be mediated by viral proteins, specifically Tat, which continues to be produced under viral suppression. Recent studies suggest that Tat may affect dopamine homeostasis by inhibiting dopamine transporter function allosterically, potentially elevating extracellular dopamine [98,99,100]. Psychostimulants, such as cocaine and meth, have complex interactions with Tat. Intravenous cocaine self-administration increased striatal DAT binding and showed an increased sensitivity to cocaine’s reinforcing effects in transgenic rats with Tat expression through a leftward shift in the dose-response curve [97]. Tat induces conformational changes in DAT that increase the affinity of cocaine for the transporter protein [101]. The similar and distinct effects of chronic cocaine administration and HIV-1 infection appear to enhance neurotoxicity, as evidenced by the overexcitation of mPFC pyramidal neurons in transgenic rats with Tat expression [102]. HIV-Tat and cocaine combined exposure in human and rat primary hippocampal neurons caused a significant depolarization of the mitochondrial membrane potential, indicative of mitochondrial damage. In contrast, Tat alone or cocaine alone only caused a slight effect on mitochondrial membrane potential [103].

Expression of Tat in transgenic mice augments methamphetamine-induced sensitization, as shown by increased locomotor activity and decreased expression of dopamine receptors demonstrated with RT-PCR and immunohistochemistry [79]. Like the mechanism of meth, Tat alters DA homeostasis by inhibiting DAT, and the combination of Tat and meth decreases DAT function more than either condition alone [104]. Increasing doses of meth resulted in impaired working and spatial memory in Tat transgenic mice compared to non-Tat or non-meth-treated mice, suggesting cooperative effects between Tat and meth on neurocognition [105]. Tat also increases microglial activation, indicating neuroinflammation [79]. Combined exposure to Tat and meth increased cellular ROS production, leading to neuronal injury [106]. In sum, these studies find that HIV infection enhances the effects of psychostimulants on dopaminergic pathways in the brain, including the reward pathway. This raises the possibility that PLWH have a greater risk of developing SUDs than those without HIV. Future studies examining compounds that can specifically block Tat binding site(s) in DAT and new forms of cART that do not affect normal DAT function, may provide an early intervention for mitigating the negative effects of HAND in PLWH with SUD.

The effects of gp120 on the dopaminergic system and drug reward are not as well characterized as those of Tat. Hu et al. 2009 were the first to report that human dopaminergic neurons exposed to gp120 decreased DA uptake significantly, caused a loss of dopaminergic neurons, and induced oxidative damage. Thus, gp120′s disruption of DAT function may further exacerbate the effects of cocaine and meth. There are limited studies investigating the combined exposure of gp120 and cocaine or meth, which we highlighted in the sections below. 

## 4. Behavioral and Neurological Characterization of Comorbid HIV and Cocaine/Methamphetamine Exposure in Tat or gp120 Transgenic Rodent Models of HIV-1

### 4.1. Tat/Cocaine

Pre-clinical studies that model HAND using Tat expression in transgenic mice and rats show synergistic and interactive effects of Tat and cocaine or meth exposure on behavior and brain structure and function. In the following section, we will review the behavioral and neurological deficits associated with combined Tat expression and cocaine or meth exposure and the sex differences between these deficits. The reviewed literature includes different drug conditioning paradigms. Thus, we will review the literature in the context of the experimental designs.

Studies that evaluated the effects of cocaine in transgenic rodents with Tat expression (Tat+/Cocaine+) provide evidence for the additive effects of cocaine exposure and Tat expression on behavioral impairments. Few studies have evaluated the effects of cocaine self-administration on behavior in rodents with Tat expression within the past ten years. One study revealed that Tat+ mice self-administered more cocaine at escalating doses than Tat- mice [97]. This evidence suggests that Tat exacerbates sensitivity to the rewarding effects of cocaine; however, further investigations using this drug administration paradigm are required. Consistent with this, studies that used a cocaine-condition place preference paradigm (cocaine-CPP) showed synergistic effects of Tat expression and cocaine exposure. Cocaine-CPP is used to measure the motivational effects of cocaine by exposing rodents to cocaine in an established environmental preference and then measuring place preference without cocaine exposure [107]. A study that used cocaine-CPP showed that Tat+ mice exhibited more cocaine-CPP and locomotor activity than Tat- mice [17]. To further investigate how Tat expression changes cocaine-CPP, researchers induced Tat expression in mice that exhibited cocaine-CPP. After Tat induction, there was a 3.1-fold increase in cocaine-CPP compared to previous place preferences [17]. Together, this evidence shows that Tat expression exacerbates cocaine-induced sensitization, conditioning, and motivation. This points to the importance of addressing CUD in people diagnosed with HAND, as they could exhibit a greater severity of CUD than someone with CUD alone.

A study using a chronic drug regimen of cocaine found interactive effects of combined Tat expression and cocaine exposure on cognition. Researchers employed behavioral tasks measuring a range of cognitive functions and observed that only Tat+ mice exposed to cocaine exhibited greater impairments in working memory than drug-naive controls [108]. There were no other significant interactive effects for behavioral tasks that tested for balance, coordination, or locomotor activity [108]. This suggests that working memory is a relatively unique deficit associated with chronic cocaine exposure in HAND; however, further investigations using other drug administration paradigms and more comprehensive behavioral testing are needed to determine the specificity of these findings more definitively. 

The exacerbated effects of Tat expression and cocaine exposure on reward, conditioning, motivation, drug sensitization, and working memory in transgenic rodents may be explained by electrophysiological dysregulation and structural abnormalities in the pre-frontal cortex (PFC), striatum, and hippocampus. The neuropathological effects of HAND and CUD affect the PFC, striatum, and hippocampus [30,82,109,110,111,112]. Thus, it is important to highlight the impact of combined Tat expression and cocaine exposure on these regions to determine future directions in investigating pharmacological interventions. A study that employed a cocaine self-administration paradigm and induced Tat expression in vivo found that Tat+/Cocaine+ and Tat-/Cocaine+ rats both exhibited increased excitation of medial PFC (mPFC) pyramidal neurons; however, Tat+/Cocaine+ rats showed the greatest excitation [20]. Interestingly, a study that employed cocaine self-administration and exposed mPFC tissue to Tat in vitro showed rats exposed and not exposed to cocaine exhibited increased excitation of mPFC pyramidal neurons, with cocaine-exposed rats showing the greatest increase [19]. Combined, these results suggest the non-temporal-specific synergistic effects of combined cocaine exposure and HAND on the hyperexcitability of the mPFC. In the striatum, combined Tat expression and cocaine self-administration reduced affinity and increased abundance of low-affinity dopamine reuptake transporters (DATs) compared to Tat+/Cocaine- rats [97]. This may be a compensatory mechanism for cocaine exposure and HIV-induced inhibition of DAT [78,113]. Dysregulation of the dopamine reuptake mechanism (through DAT) may be an underlying mechanism for the effects of Tat+/cocaine+ on reward, conditioning, motivation, and drug sensitization; however, more comprehensive studies that evaluate the relationships between DAT binding and reward functioning are needed before characterizing DAT as a therapeutic target for addressing these behavioral impairments.

Structural abnormalities in neuroimmune cells in the hippocampus may explain the interactive effects of Tat+/Cocaine+ on impaired working memory. A study that used a chronic cocaine administration regimen observed that only Tat+/Cocaine+ mice exhibited reduced branch lengths of microglia and dendritic swelling in the hippocampus [108]. These morphological deficits may be due to the effects of combined cocaine exposure and Tat expression on neuroinflammation through enhanced microglia activity [114,115].

Despite limited studies on sex differences in behavior and neurological functioning among Tat+/Cocaine+ rodents, evidence suggests estrogen may be a neuromodulator of these deficits. A study that employed the cocaine-CPP paradigm in female mice in the diestrus (low-hormone) and proestrus (high-hormone) phases of the estrogen cycle showed that diestrus mice with Tat expression exhibited greater cocaine-CPP than proestrus mice with Tat expression. Before cocaine sensitization was established, only female mice in the diestrus phase with Tat expression exhibited lower cocaine-induced locomotor activity than Tat- mice or Tat+ proestrus mice [116]. These results suggest that the acute effects of cocaine are greater in the diestrus phase than in the proestrus phase and that the estrogen cycle exerts less influence on the effects of chronic cocaine exposure. Thus, the estrous cycle may be a therapeutic target for women living with HAND at high risk for CUD, which merits investigation. Research that evaluates if cyclical sex hormone fluctuations in Tat+/Cocaine+ male rodents is needed to determine if males living with HAND require more sex-specific pharmacological therapies.

### 4.2. Tat/Meth

Tat+/Meth+ rodents exhibit poorer working memory and greater drug sensitization than Tat+/Meth- or Tat-/Meth+ rodents. Tat+ mice sensitized to meth through a chronic administration regimen exhibited greater locomotor activity than Tat- mice during the meth challenge, suggesting synergistic effects of Tat expression and chronic meth exposure on drug sensitization [79,117]. A study that employed a binge meth regimen in transgenic mice with and without Tat expression observed that only Tat+/meth+ mice exhibited poorer working memory than controls [105]. Researchers also found that Tat+/Meth+ and Tat+/Meth- mice similarly exhibited poorer spatial memory than controls [105]. These results are consistent with evidence showing that working memory impairments are specific to Tat+/Cocaine+ rather than Tat+/Cocaine- or Tat-/Cocaine+ [108]. Combined, these findings may point to working memory as a therapeutic target for comorbid HAND and psychostimulant use disorders. Future studies should directly compare working memory impairments between Tat+/Cocaine+ and Tat+/Meth+ mice to determine the generalizability of these results. Tat expression also increases the rewarding effects of meth, as measured through intracranial self-stimulation (ICSS). The ICSS procedure is an operant paradigm that pairs a lever press with an electrical shock to the medial forebrain bundle to evaluate the neurocircuitry underlying the reinforcing effects of addictive substances [118]. A study using ICSS in Tat+ and Tat- mice after administration of the binge-meth regimen observed that Tat+ mice exhibited a lower reward threshold than Tat- mice [119]. This indicates Tat+ mice showed greater sensitivity to the reinforcing effects of meth compared to Tat- mice [119]. A study that compared ICSS in Tat+ mice during withdrawal from chronic or binge meth administration regimens demonstrated that mice from both regimens exhibited higher reward thresholds than Tat-, indicating more reward deficits or anhedonia regardless of pattern of use [120]. Combined, these results provide comprehensive evidence in support of the exacerbated effects of combined Tat expression and meth exposure on the neurocircuitry underlying reward.

The deficits in working memory and exacerbated meth sensitization observed in Tat+/Meth+ transgenic rodents may be due to neurological deficits in the striatum, PFC, and hippocampus. Neurological deficits in Tat+/Meth+ transgenic rodents are observed on a molecular level, such as through dysregulation of dopamine receptor expression, autophagy markers, and growth factors. Studies that used a chronic meth administration regimen in transgenic rodents observed that Tat potentiates the neuropathological mechanism of meth exposure. A study that evaluated dopamine receptor density in Tat+/Meth+, Tat+/Meth-, Tat-/Meth+, and Tat-/Meth- mice showed Tat+/Meth+ exhibited the lowest dopamine receptor D2 (DRD2) protein expression in the caudate putamen compared to all groups [79]. These results provide evidence on the additive effects of Tat expression and chronic meth exposure on the dopamine meso-striatal system. The downregulation of DRD2 in the caudate putamen may explain the exacerbated locomotor activity in Tat+ mice chronically exposed to meth, as striatal DRD2 pathways contribute to controlling meth-induced locomotion [121,122].

Rats with Tat expression chronically exposed to meth exhibited higher autophagy markers in striatal dopamine-receptor-expressing neurons than Tat+/meth- and Tat-/meth+ rats [106]. This evidence suggests that chronic meth exposure enhances HIV neuropathogenesis. Autophagy may play a role in oxidative stress, as the administration of reactive oxygen species (ROS) scavenger N-acetylcysteine amide lowered the expression of autophagy markers in Tat+/Meth+ rats [106]. This suggests ROS scavengers may effectively reduce neuropathogenesis from HIV in people who use meth chronically. The impairments in working memory observed in transgenic mice with Tat expression administered meth binge regimen may be due to reduced brain-derived neurotrophic factor (BDNF) gene in memory-related brain regions (i.e., hippocampus, parietal cortex, PFC). A study that administered a binge meth or saline regimen to transgenic mice with and without Tat expression observed that Tat+/Meth+ mice exhibited reduced levels of BDNF in the parietal cortex, hippocampus, PFC, and cerebellum. In contrast, reduction in BDNF was only observed in the parietal cortex and PFC in Tat+/Meth- and Tat-/Meth+ mice [105]. This shows that Tat expression and binge meth exposure have more pervasive adverse effects on the brain. Reduction in BDNF is often associated with working memory impairments in rodent models of psychiatric conditions such as depression and schizophrenia [123,124,125]. Recent evidence shows that aerobic exercise increases BDNF levels and improves working memory in rodents [126,127]. This suggests that lifestyle interventions may help mitigate working memory impairments in people living with comorbid HAND and MUD.

Estrogen may play a role in the oxidative stress mechanism of Tat protein expression and binge meth administration in transgenic mice. When researchers administered ROS scavenger N-acetylcysteine amide to male and female mice with and without Tat expression following a meth challenge, they only observed attenuation of meth-induced locomotion and increases in DRD2 expression in Tat- females and males and in Tat+ males, but not in Tat+ females [117]. The blunted response to the ROS scavenger in Tat+ females is likely due to changes in estrogen levels in the brain, which alters redox homeostasis [128]. Also, Tat expression and chronic meth exposure exacerbate the reduction in protein expression responsible for redox balance compared to Tat+/Meth- and Tat-/Meth+ [106]. In humans, when estrogen levels decrease, especially in adult females, there is a decrease in energy production and an increase in brain ROS levels [128]. Future directions should evaluate the interactive effects of age and estrogen levels in Tat+/Meth+ female rodents to investigate more sex-specific pharmaceutical interventions.

### 4.3. Gp120/Cocaine

Like Tat, gp120 expression is used to model HAND in transgenic rodent models; however, it is important to note that the majority of the gp120 studies that met the criteria for this review primarily focused on the combined effects of gp120 expression and meth exposure on neuroimmune and behavioral functioning, with limited characterization of the effects of gp120+/Cocaine+. Below, we review the behavioral, neurological, and sex differences observed in gp120+/Cocaine+ and gp120+/Meth+ transgenic rodents, with a primary focus on gp120+/Meth+ studies, as they have predominated the literature within the past ten years.

Abstinence from chronic cocaine self-administration and gp120 protein expression decreases responsiveness to pharmacological interventions. MC-25-41, a weak D3R partial agonist, has been shown to reduce cocaine motivation in rats; however, in a study that administered MC-25-41 to rats with and without gp120 expression, reduced cue-induced cocaine seeking occurred only in rats without gp120 expression [80]. This evidence suggests that HAND may reduce treatment efficacy for cocaine seeking, highlighting the importance of investigating novel therapeutic interventions for people living with HAND and CUD.

Combined gp120 and cocaine exposure induce neurological dysfunction through neuroimmune cell energy deficits. An in vitro study using CHME-5 microglial cells and primary astrocytes transformed from rats showed how combined gp120 expression and cocaine exposure exacerbated oxidative stress [129,130]. Researchers found that gp120+/Cocaine+ cells exhibited greater ATP utilization than controls, while gp120+/Cocaine- did not. Further, gp120+/Cocaine+ and gp120-/Cocaine+ exhibited increased oxygen consumption than controls, with gp120+/Cocaine+ rats exhibiting the highest oxygen consumption rates [130]. Additionally, all groups exhibited greater levels of ROS than controls, with gp120+/Cocaine+ showing the highest levels [130]. This evidence suggests that gp120+/Cocaine+ exposure exacerbates neurotoxicity compared to gp120 expression and cocaine exposure alone. Future in vivo studies are needed to determine how cocaine exposure impacts the energy consumption of neuroimmune cells in various brain regions. Such future investigations will permit researchers to identify region-specific neuroimmune cell targets.

We did not find studies that investigated sex differences or effects of sex hormones in behavior or neurological structure and functioning in gp120+/Cocaine+ rodents that met the criteria for this review. Future studies need to investigate this because of evidence demonstrating estrogen’s role in behavioral and neurological deficits in Tat+/Cocaine+ mice [116]. Expanding this field to include gp120 would provide insights into similar and distinct roles of Tat and gp120 protein expression in sex differences of behavior and brain structure and function. This will inform how scientists will investigate and develop more symptom- and sex-specific therapies, as Tat and gp120 have similar and different mechanisms in the neuropathogenesis and neurotoxic effects of HIV [81].

### 4.4. Gp120/Meth

Rodents with combined gp120 expression and binge meth exposure exhibit deficits in working memory, learning, and spatial memory. A study used the Barnes Maze test to evaluate spatial memory by calculating the time mice spent between the target and other quadrants. While gp120+/Meth-, gp120-/Meth+, and gp120+/Meth+ mice exhibited a smaller time difference between the target and other quadrants than controls, gp120+/Meth+ mice exhibited the smallest time difference [131]. These results suggest that combined gp120 protein expression and binge meth exposure worsen working memory impairments. Another study that used the Barnes Maze test to measure learning based on the percentage of errors before finding the target quadrant reported that gp120+/Meth+ mice exhibited lower scores than gp120+/Meth-, gp120+/Meth-, and controls on the last day of acquisition [132]. This may indicate that gp120+/Meth+ induces more learning impairments than gp120 expression or binge meth administration alone. A study used the attentional-set-shifting test, which involves researchers presenting textures or odors paired with a food pellet and introducing an interfering stimulus to test for discrimination learning [133]. Results showed that gp120+/Meth+ mice exhibited the greatest failure rates in visual discrimination, suggesting that combined gp120 expression and binge meth exposure exacerbate learning deficits [133]. Future studies should evaluate learning and memory processes in meth self-administration and chronic meth exposure paradigms to understand how these findings translate to people living with comorbid HAND and MUD.

The learning and memory impairments observed in gp120+/Meth+ mice may be explained by electrophysiological impairments in the hippocampus. A study that acutely exposed rat hippocampal slices to gp120 and meth independently showed reduced synaptic transmission in the CA1 region and reduced long-term potentiation in the CA3-1 regions [134]. Co-treatment of hippocampal slices with gp120 and meth exacerbated these effects, suggesting synergetic effects of gp120 and acute meth exposure [134]. Combined binge administration of meth and gp120 expression reduced post-tetanic potentiation in the hippocampus compared to controls, while gp120+/Meth+ did not exhibit differences from controls [131]. Together, this evidence suggests that the electrophysiology of the hippocampus may be a therapeutic target for the learning and memory impairments observed in gp120+/meth+ mice [132,133]. Future studies should examine neurotransmitter receptor expression to determine the molecular mechanisms underlying electrophysiological dysregulation of the hippocampus in rodents administered meth in vivo. This will allow clinicians and scientists to investigate and identify biochemical therapeutic targets in co-occurring HAND and MUD.

Sex differences in the effects of gp120 expression and meth exposure on the brain and behavior within the past ten years are limited. Two studies provided evidence for age and sex interactions in sensorimotor gating as measured by pre-pulse inhibition (PPI), which is a startle reduction to a strong auditory stimulus preceded by a weak auditory stimulus [135]. One study investigating PPI in mice withdrawn from binge meth exposure did not observe any differences between gp120+/Meth+ male and female mice [136]. Another study that measured PPI in male and female mice at 8, 14, and 22 months showed that only aged male gp120+/Meth+ mice exhibited reduced PPI, providing evidence for impaired sensorimotor gating [137]. These results suggest that older men with comorbid HAND and MUD may exhibit impairments in information processing. The result from this study also highlights the importance of early HIV treatment in men with MUD, as they could suffer from more severe cognitive impairments. Future studies should investigate the interactions between gp120, meth exposure, and sex hormones in the brain regions underlying PPI to characterize the mechanisms underlying these sex differences.

## 5. Behavioral and Neurological Characterization of Comorbid HIV and Cocaine Use Disorder/Methamphetamine Use Disorder from Clinical Evidence

### 5.1. HIV/CUD

Clinical studies have explored the effects of CUD on behavior and brain structure and function among PLWH through cognitive battery assessments, structural MRI, resting-state fMRI, task-based fMRI, and PET imaging. Studies that used monetary decision-making tasks revealed HIV+/CUD+ exhibited more risky decision-making than HIV+/CUD- and HIV-/CUD+ [94,138]. This impaired decision-making may be related to inhibitory control deficits associated with HIV+/CUD+ that are persistent through abstinence from chronic cocaine use. HIV+/CUD+ exhibited lower correct response rates on a Go/No-Go task than HIV+/CUD- and HIV-/CUD+, indicating impaired response inhibition [139]. Consistent with the exacerbated behavioral impairments observed in transgenic rodent models of HAND with cocaine exposure, HIV+/CUD-, HIV-/CUD+, and HIV+/CUD+ exhibited poorer decision-making and inhibitory behavior than controls, with HIV+/CUD+ showing the most impairments [17,94,108,138,139]. This provides evidence that HIV increases vulnerability to behavioral impairments associated with CUD, pointing to the importance of addressing this comorbid condition.

Task-based fMRI studies have highlighted the neurocircuitry underlying these behavioral impairments. When making decisions about monetary rewards, HIV+/CUD+ exhibited heightened global activation of the PFC during easy choices and lower activation during hard choices. In contrast, HIV+/CUD- and HIV-/CUD+ only exhibited increased activation in the PFC during hard choices [138]. Meade et al. further administered an economic loss aversion fMRI task; HIV+/CUD+ exhibited economic-gain-related activation in the vmPFC and posterior precuneus and economic loss-related activation in the anterior cingulate cortex [94].

Studies that used comprehensive neuropsychological batteries in the cART era did not show differences in cognitive functioning between HIV+/CUD+, HIV+/CUD-, HIV-/CUD+, and HIV-/CUD- [139,140,141]; however, other studies revealed differences in activation patterns of brain regions associated with neurocognitive functions, suggesting the effects of HIV+/CUD+ may affect neurocognitive functioning at subclinical levels. As risk increased in a probabilistic monetary decision-making fMRI task, HIV+/CUD+ showed heightened activation of the hippocampus, post-central gyrus, lateral occipital cortex, cerebellum, and posterior parietal cortex. In contrast, HIV+/CUD-, HIV-/CUD+, and HIV-/CUD- exhibited lower activation [141]. Hyperactivation may be a compensatory mechanism for the neurodegenerative effects of HIV+/CUD+, similar to that observed in transgenic rodent models of HAND with cocaine exposure [19,20,142]. Researchers recently evaluated the effects of HIV+/CUD+ on brain glucose metabolism using PET with an 18F-FDG radiotracer. Their results showed that HIV+/CUD+ had the lowest levels of 18F-FDG uptake globally, while HIV+/CUD-, HIV-/CUD+, and HIV-/CUD- exhibited moderate to high levels [143]. These results may explain the altered brain activation patterns observed in the fMRI studies, as glucose metabolism is a primary energy source for neurons [144,145]. Neurological energy deficits are observed in gp120+/Cocaine+ rodent-derived primary astrocytes through greater ATP utilization than gp120-/Cocaine+ and controls [130]. Future clinical studies should evaluate energy deficits in neuroimmune cells to determine the mechanism underlying the low 18F-FDG uptake observed in HIV+/CUD+. There are comparatively few studies on the effects of HIV+/CUD+ on brain structure. A recent diffusion tensor imaging study in HIV+/CUD+ people who were abstinent from cocaine use revealed a main effect of HIV on globally reduced fractional anisotropy, a marker of white matter integrity; however, they found no significant interaction between HIV+ and CUD+ [146]. The effects of CUD on brain structure may be too small to detect the combined effects of HIV and CUD, but further investigations in abstinent and active HIV+/CUD+ groups are needed to corroborate this.

Multi-site epidemiological evidence in the cART era shows men living with HIV are more likely to exhibit SUDs and poorer adherence to treatment than women living with HIV [5,147,148]. There are few clinical studies investigating sex differences in neurological and behavioral function in HIV+/CUD+, though recently some studies have examined sex-dependent effects associated with HIV and CUD independently. In PLWH, women have poorer executive functioning, fine motor skills, and psychomotor speed than men [149,150]. Consistent with this, a machine learning study that investigated sex-dependent cognitive profiles in HIV revealed women living with HIV had poor motor function, learning, and delayed recall. In contrast, men living with HIV exhibited strengths in learning and processing relative to women living with HIV [151]. The sex differences in CUD manifest in differential symptomology and responses to treatment. Women with CUD are more likely to experience drug cravings, guilt after drug use, psychiatric comorbidities, and sensitivity to drug cues than men [152,153]. Randomized clinical trials that tested the efficacy of disulfiram and guanfacine, medications that have been studied as possible CUD treatments, revealed sex-specific outcomes. Disulfiram, a dopamine beta-hydroxylase inhibitor, is more effective in men than women, with men exhibiting more days of abstinence from cocaine use during treatment [154,155]. Conversely, guanfacine, a norepinephrine inhibitor, is more effective in women than men, which may be due to guanfacine’s ability to attenuate the peripheral sympathetic stress system, targeting sex-specific symptoms observed in women [156,157,158]. The sex differences independently observed in HIV and CUD suggest they may influence outcomes in a sex-dependent manner. Consistent with this, pre-clinical evidence suggests estrogen impacts the acute effects of cocaine, pointing to a target for the risk of CUD in women living with HIV [116]. Future clinical studies should evaluate HIV+/CUD+ sex differences in behavior, cognition, neurological function, and the role of sex hormones to inform clinicians about sex-specific behavioral and pharmaceutical interventions.

Epidemiological evidence shows NHB PLWH are more likely to exhibit severe HAND symptoms and frequent cocaine use than White PLWH [159,160,161], which likely reflects a greater accumulation of adverse social determinants of health among NHB populations [162]. There are few studies on the ancestry effects of HIV+/CUD+ on neurological and behavioral function, highlighting the need for more racially diverse samples. Recent evidence suggests HIV+/CUD+ may differentially impact NHB PLWH due to the social barriers to receiving HIV prevention care and treatment, especially in rural NHB communities. Rural NHB who use cocaine are less likely to perceive HIV testing as acceptable regardless of testing site type (i.e., community health center, mobile van, physician) than NHB people in urban areas who use cocaine [163]. Among rural NHB people who use cocaine, social unacceptability was the only barrier to accessing HIV testing, while affordability, geographic availability, and physical accessibility were not [164]. An evaluation of social determinants to seeking HIV testing among rural NHB people who use cocaine revealed those who are women, young adults, or frequently receive testing for sexually transmitted diseases predicted testing for HIV [165]. This points to specific sub-groups among rural NHB who use cocaine who are likely not to seek HIV testing. These community-based studies highlight the need for culturally informed interventions to engage NHB Americans in rural areas in HIV prevention care to lower the incidence and prevalence of HIV+/CUD+.

### 5.2. HIV/MUD

Clinical studies have examined the effects of HIV+/MUD+ on neurocognition and brain structure and function using cognitive batteries, structural MRI, postmortem tissue analyses, and social genomics. It is unclear how HIV+/MUD+ impacts cognitive functioning compared to either condition alone. A recent study observed that HIV+/MUD+ had greater self-reported emotion dysregulation than HIV+/MUD- [124]. Conversely, studies that evaluated other cognitive functions such as sustained attention, vigilance, impulsivity, and emotion recognition did not find additive effects of HIV+/MUD+ [166,167,168]. This suggests that emotion regulation represents a possible specific behavioral therapy target for HIV+/MUD+; however, it remains unclear how specific emotion regulation domains (i.e., cognitive reappraisal, emotional awareness) are impaired. This evidence may also point to other aversive mediating effects of HIV+/MUD+ since cognitive impairments were observed in HIV+/MUD- and HIV-/MUD+ [166,167,168]. Impairments in emotion regulation observed in HIV+/MUD+ are consistent with the hypodopaminergia associated with these comorbid conditions since dopamine plays a key role in cognitive control over emotions [169,170].

Like HIV+/CUD+, few studies have examined sex differences in the effects of HIV+/MUD+ on behavior and brain structure and function. As discussed above, women living with HIV have poorer cognitive functioning than men living with HIV [149,150,151]. A study that investigated sex differences in impulsivity and brain structure associated with MUD revealed that smaller frontal cortical volumes were associated with greater impulsivity only in women [171]. Also consistent with sex differences in CUD, women with MUD experience poorer outcomes relative to MUD men. Women in treatment for MUD are more likely to exhibit psychological burden, emotion dysregulation, and childhood and sexual trauma than men in treatment for MUD [172]. This shows the need for more comprehensive gender-specific biopsychosocial treatments to close this health disparity gap.

There is limited knowledge on the effects of HIV+/MUD+ on brain structure or function relative to either condition alone. HIV+/MUD+ exhibited larger global cortical areas and volumes in the precentral and paracentral gyri than HIV+/MUD-, HIV-/MUD+, and HIV-/MUD-, but these results did not survive correction for multiple comparisons [173]. Researchers suggest that the effects of MUD on the brain among PLWH are too subtle to detect using current neuroimaging methods. On a molecular level, HIV+/MUD+ exhibit higher levels of mitochondrial DNA (mtDNA) deletions in gray matter tissue than HIV+/MUD-, HIV-/MUD+, and HIV-/MUD- [18]. The effects of these mtDNA deletions on the brain manifest through global impairments of cognitive function in HIV+/MUD+ [18]. The accumulation of mtDNA deletions contributes to mitochondrial dysfunction and an overall energy reduction and is typically associated with neurodegeneration [174,175]. These mtDNA deletions may be explained by the neuropathological mechanism of mitochondrial ROS produced in HIV+/MUD+ [176]. Environmental stress worsens the molecular neuropathological effects of HIV+/MUD+. A postmortem brain study investigated this by analyzing epigenetic changes associated with HIV+/MUD+. Researchers found that HIV+/MUD+ exhibited the highest levels of global DNA methylation and expression of genes related to DNA methylation in the frontal cortex [177]. DNA methylation is associated with accelerated brain aging in HIV+, and HIV+/MUD+ may exacerbate these effects [3,178,179].

Over the past ten years, a growing body of literature has demonstrated that HIV+/MUD+ in MSM exhibit poorer health outcomes than heterosexual HIV+/MUD+ [14,180]. Social adversity may exacerbate the epigenetic changes observed in HIV+/MUD+ MSM. Greater experience of stress related to being a sexual minority was associated with accelerated epigenetic aging, evident through telomere shortening and genome-wide DNA methylation patterns in the brain [16]. These results provide evidence for the molecular manifestations of the interactive relationships between social and biological mechanisms underlying HIV+/MUD+, consistent with Dr. Wakim-Takaki’s biopsychosocial model. Treating this dual diagnosis in MSM is difficult as they exhibit poor adherence to cART and limited social functioning [34,35]. This points to the importance of population-specific medical and behavioral treatments for HIV+/MUD+ to improve health equity.

## 6. Discussion and Conclusions

This review addressed the neuropathogenesis, behavioral, and neurological impairments of comorbid HIV and psychostimulant use disorders (CUD and MUD) per Dr.Wakim-Takaki’s biopsychosocial model [15]. The evidence reviewed supports the dopaminergic dysregulation implicated in comorbid HIV and psychostimulant use disorders; however, the exact mechanisms by which HIV and dopamine interact to contribute to HAND in PLWH with SUD require further investigation. Future studies should combine results from various modalities to better understand the impact of drug-induced changes in dopaminergic systems involved in HIV. Dopamine dysregulation may underlie the drug hypersensitization and cognitive deficits observed in rodent models of HAND with meth or cocaine exposure [17,19,20,79,80,105,108,119,120,132,133,181]. There is limited characterization of the cognitive deficits associated with HIV+/CUD+ and HIV+/MUD+. The majority of the reviewed studies that used comprehensive cognitive batteries did not find additive or interactive effects of HIV+/CUD+ or HIV+/MUD+ [139,140,141,166,167,168]. Results from clinical studies suggest the combined effects of these comorbid conditions may not manifest on a cognitive level. These results may differ from transgenic rodent models of HAND because only one HIV protein was expressed, while numerous HIV proteins are expressed in humans. Alternatively, this evidence may point to the need to explore more specific cognitive processes. One study reviewed found HIV+/MUD+ to be associated with emotion dysregulation; however, emotion regulation was not examined in HIV+/CUD+ [124]. Researchers should explore emotion regulation processes in HIV+/CUD+ and further in HIV+/MUD+, as it is a plausible transdiagnostic risk factor for neuropsychiatric conditions [182,183,184].

While HIV+/CUD+ and HIV+/MUD+ have similar complex relationships between their biological, behavioral, and social mechanisms, the mechanisms investigated have differed. Pre-clinical and clinical studies on the combined effects of HIV and cocaine exposure have focused on impairments in reward function, decision-making, and altered structure and function in dopaminergic brain regions [17,19,20,80,94,108,138,139,143]. On the other hand, pre-clinical and clinical evidence on the combined effects of HIV and meth exposure have focused on molecular-based effects, such as dysregulation in the expression of cellular signaling proteins and mitochondrial dysfunction [18,105,106,131,134,177]. Future neuroimaging studies should evaluate the effects of HIV+/MUD+ on neurological function using task-based fMRI and PET imaging to provide a more detailed characterization of the neurological effects of this comorbid condition.

Pre-clinical evidence demonstrated sex differences in behavioral and neurological impairments associated with HIV and cocaine or meth exposure [106,116,117,137]; however, the clinical evidence has only evaluated sex differences in HIV and psychostimulant use disorders independently, likely due to limitations in sample sizes. Researchers should investigate sex differences between HIV+/CUD+ and HIV+/MUD+ in a community-based sample, as social adversity exacerbates their effects in a population-specific manner [16,163,164,165]. Additionally, researchers should evaluate levels of estrogen in women living with HIV and a psychostimulant use disorder, as pre-clinical evidence suggests estrogen may be a biochemical target for mitigating the effects of HIV and psychostimulant exposure on drug sensitization and responsiveness to ROS scavenger treatment [106,116,117]. One study found that aged male gp120+/Meth+ had more impairment in sensorimotor gating compared to all groups, while gp120+/Meth- and gp120-/Meth+ male and female mice did not exhibit significant differences from controls [137]. This points to the importance of evaluating interactions between sex and age in people diagnosed with HAND and MUD, as they could exhibit unique deficits compared to either conditional alone.

HIV+/CUD+ differentially impacts NHB people because of the stigma associated with HIV testing, especially in NHB rural communities [163,164,165]. This may be due to deep-rooted homophobia and medical mistrust within NHB culture [185,186,187]. Researchers have introduced faith-based interventions centered around NHB American culture to combat this. Studies that used these interventions in rural settings found reductions in individual-level stigma [188,189]. To determine if these results are community- or race-specific, Derose et al. integrated HIV education workshops, congregation-based HIV testing, and sermons catered to urban NHB and Latino communities. They found both Latinos and NHB people exhibited increases in HIV testing; however, there were evident disparities among these groups, with NHB reporting more medical mistrust and stigma toward HIV testing [190]. Together, these findings suggest that comprehensive church-based interventions catered to NHB may reduce the social barriers to medical care.

MSM continue to be the largest risk group for HIV infections in the U.S., where crystal meth misuse heightens the risk for HIV infection through greater engagement in condomless anal sex [10]. HIV+/MUD+ MSM exhibit epigenetic changes associated with the social stress of being a sexual minority; however, future studies should investigate this in other sexual minority groups to determine the specificity of these findings. To combat this social stress, researchers have used an emotion regulation-focused intervention that targets trauma related to HIV infection. They found HIV+/MUD+ MSM significantly reduced their meth use [191]. A study that used behavioral activation, an intervention aimed at targeting pleasure and goal-directed behaviors in HIV+/MUD+ MSM, found lower engagement in condomless anal sex and longer abstinences from sex and meth use [192]. Together, these interventions demonstrate the efficacy of behavioral community-based interventions in lowering the likelihood of HIV transmission and the persistence of chronic meth use in MSM. The effects of social adversity and the efficacy of interventions in NHB PLWH and MSM living with HIV support Dr. Wakim-Takaki’s biopsychosocial model, suggesting the need to address this dual diagnosis in a population-centered manner by addressing social, behavioral, and neurological mechanisms [15]. Machine learning is becoming a useful tool to identify biological, behavioral, and social determinants of SUD and HIV independently; however, to our knowledge, it has not been applied to identify predictors of comorbid HIV and SUD [193,194,195,196]. Data-driven models that predict determinants of comorbid HIV and SUDs will inform researchers and clinicians on developing more comprehensive diagnostic testing and population- and patient-centered care. More comprehensive pre-clinical and community-specific investigations and interventions will sustainably improve patients’ quality of life, as people living with these comorbid conditions exhibit unique biological, psychological, and sociological challenges that need to be addressed holistically.

## Data Availability

Not applicable.

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
