# Peer review of "Neurological, Behavioral, and Pathophysiological Characterization of the Co-Occurrence of Substance Use and HIV: A Narrative Review"

_brainsci, 2023, doi:10.3390/brainsci13101480_

Round 1
Reviewer 1 Report (Previous Reviewer 2)
This is a highly informative and well organized review on an important topic. Except for a few typos (e.g., p. 3, line 152), this manuscript is ready for publication. The authors have done an excellent job responding to comments and restructuring the parts requested.
See above - only minor edits needed
Author Response
Please see the attachment

Reviewer 2 Report (Previous Reviewer 3)
It’s important to clearly indicate in the title and throughout the text that it is indeed a narrative review and not a systematic review.
By including "Narrative Review" in the title, you make it clear to your audience that your review is not following the rigorous systematic review methodology but rather providing a comprehensive overview and synthesis of existing literature on the topic.
Author Response
Please see the attachment

This manuscript is a resubmission of an earlier submission. The following is a list of the peer review reports and author responses from that submission.
Round 1
Reviewer 1 Report
There are substantial issues with the organizational structure, language, and scientific rigor of the preclinical section of this manuscript. Below I highlight several major issues with the manuscript and other specific comments I had that all contribute to my overall opinion.
Major comments - these comments represent commentary about the manuscript as a whole that are largely structural in nature:
-The abstract should be amended to better reflect the narrow focus of the preclinical half of this review. Currently, the abstract reads as if the review is going to be a comprehensive assessment of HIV interactions with cocaine and methamphetamine on dopamine systems, cognition/behavior etc. However, only Tat transgenic rodent models are discussed. This is a *very* narrow slice of the preclinical HIV/SUD literature. This isn't necessarily a problem so long as the paper makes is explicitly clear in places like the abstract, title, etc. that this is the intended goal.
-Related to the previous comment, it is largely unclear why the author's chose to limit their preclinical literature search to only studies using Tat models. There are other (transgenic) models (e.g., HIV-1 Tg rat, gp120 Tg mice, SIV-infected NHPs, etc) that exhibit differential and converging findings within different models of SUDs (e.g., CPP, self-administration, etc.). Moreover, one of the goals of this paper seems to be to bridge the knowledge we have from preclinical studies with observations from clinical investigations of humans who are infected with a virus that expresses many more proteins than just Tat (e.g., gp120, Nef, etc.) Again, it's not necessarily a problem if the authors want to focus on Tat effects, but a reasonable rationale for this needs to be built out early in the manuscript so that reader understands and appreciates the (narrow) focus of the paper.
-Line 89: The wording of this sentence is problematic. To say about MSM that "this population exhibits relatively poor adherence to antiretroviral therapy" without both proper and specific context is stigmatizing language. It is true that MSM, and specific subpopulations of MSM in particular, do exhibit challenges with ART adherence. However, there are specific psychosocial and environmental factors that contribute to this, which likely includes differing rates of substance use across subpopulations. When talking about marginalized and vulnerable populations such as MSM, more care needs to be taken to use very specific and scientifically-accurate language. Vague generalizations such as "this population exhibits relatively poor adherence to antiretroviral therapy" without further context is not scientifically rigorous. Please revise this sentence and any sentences like it throughout the manuscript to better reflect the cited literature.
-Related to the previous comment, there are several instances where the authors can be more specific in their language to give crucial context surrounding specific observations from the literature. Specifically, the route of administration of drug and/or the model should be noted in one way or another where appropriate. We know very well that, for example, self-administered cocaine/meth can produced overlapping yet distinct neurobiological impairments within the reward system. This very likely impacts how drug/HIV interactions are observed in preclinical models. As one of the goals of this paper is to bridge the preclinical literature with the clinical, this becomes especially important. There are a couple of examples of this pointed out in the "Specific Comments" section below.
-There is a lack of clear, organized, and parallel structure thorughout the manuscript. For example, Section 4 begins by discussing cocaine studies, with a very heavy emphasis on estrous/female hormone contributions to HIV/drug interactions within ther reward system, and then jumps into meth studies with no mention of estrous or sex differences. Overall, the manuscript is very hard to follow, with cases of literature being miscited all throughout.
Specific comments:
-Lines 41-42: The authors claim that CUD and MUD are the "most common SUDs associated with detrimental neurocognitive function among PLWH" and cite Hartzler et al., 2017 and Shiau et al., 2017. These citations do not support this claim. E.g., Hartzler et al., 2017 reports higher aggregate rates of cannabis and alcohol use disorders compared to cocaine and meth. Please amend this statement (and any statements like it in the manuscript) with appropriate citations or alter the language to more accurately reflect the literature cited.
-Define the abbreviation of methamphetamine as "meth" (e.g., lines 51-52) somewhere earlier in the text (I think the first definition of this is Line 155).
-Line 146: When introducing a new model, such as the HIV-1 Tg rat, a short description of what this model is and how it relates to the paper's focus on Tat would be helpful. In this particular example, saying something like "In an HIV-1 transgenic rat model, which constitutively and systemically expresses Tat among 6 other HIV-1 genes, ...". This will help keep the reader's focus on Tat. That said, this sentence is also a bit vague - what is meant by "striatal dysfunction"? Please be more specific with the language here and elsewhere. Also, if models other than Tat-only Tg models are going to be mentioned, effort should be made to demonstrate what effects in these models (e.g., effects on "striatal dysfunction") are likely attributable to Tat specifically over other proteins being intentionally excluded from this review (like gp120).
-Line 151: What is meant by "HIV-1 Tat and cocaine combined"? Is this an in vitro study? A Tat-tg model? Is cocaine being administered by an experimenter? Line 155 has a similar problem - what is meant by "increases sensitivity to meth reward"? Describe in more details what these studies are actually doing and what their findings are, then give us the big picture of what it all means in the context of the rest of the literature. These are really crucial details that provide necessary context. Please be more specific here and elsewhere in the text.
-Do not refer to mice as "CUD+" (e.g., Line 176). Mice do not, in any experimental scenario, have CUD.
-Lines 197-200: Do not say use terms like "HIV-1 infection" when describing transgenic mice that are expressing HIV-1 proteins. Also, McLaurin et al., 2018 utilizes the transgenic rat model (Reid et al., 2001), not mice.
Reviewer 2 Report
This is an important and interesting paper that can be used to inform work being done across multiple disciplines. There are just a few items that could strengthen this paper:'
(1) Overall, there should be careful attention to the implication of temporal associations across variables since in many cases there may be reciprocal or at least very complex indirect relationships that exist. So, while it is tempting to suggest or simply not address causation, it should be mentioned as a caveat where appropriate in sections 1-5 and particularly section 5 as well as the final section (Discussion/Conclusion).
(2) Although there may be space limitations that prohibit this, it would be helpful to provide a visual diagram of variables based on the biopsychosocial model so that readers can better grasp how the variables may interact.
(3) The review of the existing literature across domains is exceptional but the final take home message should be more emphatic and clear.
(4) In the Discussion/Conclusion section, line 454, 'however' should be preceded by a semi-colon, and any other places where that word is used as a device. Otherwise, this paper is cleanly written throughout.
See above
Reviewer 3 Report
Title
The title of the manuscript "Neurological, behavioral, and pathophysiological characterization of the co-occurrence of substance use and HIV" lacks an indication that it is a systematic review.
1. Introduction
Page 1, Line 34-35: “Before the introduction of cART, PLWH exhibited severe HIV-associated neurocog- 34 nitive disorders (HAND) and accelerated brain aging, with more prominence during the 35 late phases of HIV progression”. Please mention appropriate reference for this statement.
Page 1, line 39-40: “Compared to the general population, PLWH show a significantly higher prevalence of Sud”. Please provide the prevalence rates.
Page 2 line 73-78 “In this review, we 73 will address the effects of comorbid HIV and psychostimulant use disorders (HIV+/CUD+ 74 and HIV+/MUD+) by outlining the neuropathogenesis and dopamine dysregulation hy- 75 pothesis, characterizing neurological and behavioral impairments from pre-clinical and 76 clinical evidence, and proposing patient and population level interventions per Wakim’s 77 biopsychosocial mode”. Kindly furnish a precise and comprehensive delineation of all the objectives that the review aims to encompass. Express these objectives using a pertinent framework of question formulation.
If there exist other systematic reviews that have already addressed the same inquiry, it is pertinent to elucidate the rationale behind undertaking the present review. In instances where this review constitutes a contemporary iteration or duplication of a specific prior systematic review, it is imperative to explicitly denote this connection and provide a proper citation for the preceding review. Additionally, it is crucial to expound upon the significance underpinning the pursuit of this review. By doing so, the pivotal reasons for its undertaking can be effectively communicated.
2. Methods
The methodology employed in the article fell short in terms of its suitability for a systematic review. It failed to adequately outline the study characteristics that were employed to determine the eligibility of studies for inclusion in the review. Furthermore, explicit clarification should be provided for instances where studies were excluded due to either the non-measurement of outcomes of interest or the absence of reported results associated with these outcomes. In terms of the search strategy, authors should present a detailed, line-by-line account of the search strategy executed in each database. The complete sequence of terms used for the search should be provided. Moreover, it is crucial to present a well-structured list of outcome domains. Each outcome should be precisely defined to avoid ambiguity and facilitate a clear understanding of the variables being measured. By implementing these enhancements to the article's methodology, the systematic review will exhibit greater rigor, transparency, and clarity. This, in turn, will bolster the credibility of the findings and contribute to the overall quality of the research.
Result
Authors should include a comprehensive flow diagram illustrating various stages of the review process. It is advisable to present the key characteristics of each individual study in a tabular or figure format. This format should be designed to facilitate easy comparison of characteristics across all the included studies.
Overall, the methodology utilized in the article was inadequate in terms of its appropriateness for a systematic review. Substantial revisions are required in the manuscript, particularly pertaining to the methodology section.
Title
The title of the manuscript "Neurological, behavioral, and pathophysiological characterization of the co-occurrence of substance use and HIV" lacks an indication that it is a systematic review.
1. Introduction
Page 1, Line 34-35: “Before the introduction of cART, PLWH exhibited severe HIV-associated neurocog- 34 nitive disorders (HAND) and accelerated brain aging, with more prominence during the 35 late phases of HIV progression”. Please mention appropriate reference for this statement.
Page 1, line 39-40: “Compared to the general population, PLWH show a significantly higher prevalence of Sud”. Please provide the prevalence rates.
Page 2 line 73-78 “In this review, we 73 will address the effects of comorbid HIV and psychostimulant use disorders (HIV+/CUD+ 74 and HIV+/MUD+) by outlining the neuropathogenesis and dopamine dysregulation hy- 75 pothesis, characterizing neurological and behavioral impairments from pre-clinical and 76 clinical evidence, and proposing patient and population level interventions per Wakim’s 77 biopsychosocial mode”. Kindly furnish a precise and comprehensive delineation of all the objectives that the review aims to encompass. Express these objectives using a pertinent framework of question formulation.
If there exist other systematic reviews that have already addressed the same inquiry, it is pertinent to elucidate the rationale behind undertaking the present review. In instances where this review constitutes a contemporary iteration or duplication of a specific prior systematic review, it is imperative to explicitly denote this connection and provide a proper citation for the preceding review. Additionally, it is crucial to expound upon the significance underpinning the pursuit of this review. By doing so, the pivotal reasons for its undertaking can be effectively communicated.
2. Methods
The methodology employed in the article fell short in terms of its suitability for a systematic review. It failed to adequately outline the study characteristics that were employed to determine the eligibility of studies for inclusion in the review. Furthermore, explicit clarification should be provided for instances where studies were excluded due to either the non-measurement of outcomes of interest or the absence of reported results associated with these outcomes. In terms of the search strategy, authors should present a detailed, line-by-line account of the search strategy executed in each database. The complete sequence of terms used for the search should be provided. Moreover, it is crucial to present a well-structured list of outcome domains. Each outcome should be precisely defined to avoid ambiguity and facilitate a clear understanding of the variables being measured. By implementing these enhancements to the article's methodology, the systematic review will exhibit greater rigor, transparency, and clarity. This, in turn, will bolster the credibility of the findings and contribute to the overall quality of the research.
Result
Authors should include a comprehensive flow diagram illustrating various stages of the review process. It is advisable to present the key characteristics of each individual study in a tabular or figure format. This format should be designed to facilitate easy comparison of characteristics across all the included studies.
Overall, the methodology utilized in the article was inadequate in terms of its appropriateness for a systematic review. Substantial revisions are required in the manuscript, particularly pertaining to the methodology section.